# A Privacy-Preserving Approach to Health Insurance Fraud Detection Using Vertical Federated Learning

**DOI:** 10.3390/s25237354

**Published:** 2025-12-03

**Authors:** Raghi K R, Arjun Paramarthalingam, Harini Kanthan, Mahalakshmi Karthiban

**Affiliations:** 1Computer Science and Engineering, Sathyabama Institute of Science and Technology, Chennai 600119, TN, India; 2Computer Science and Engineering, University College of Engineering Villupuram, Villupuram 605103, TN, India; arjun@aucev.edu.in (A.P.);

**Keywords:** privacy preserving fraud detection, secure data sharing, hybrid neural network model, vertical federated learning

## Abstract

In fraud detection, centralized approaches often face challenges related to data protection, security, and potential data breaches. Such methods require sensitive healthcare and insurance data to be pooled in one location, which increases vulnerability to misuse. This paper introduces FraudNetX, a privacy-preserving fraud detection framework, by utilizing Vertical Federated Learning (VFL) to address centralized system limitations. VFL enables models to be trained collaboratively while ensuring data privacy and security through quantifiable Differential Privacy (DP) guarantees (ε = 1.0, δ = 1 × 10^5^). FraudNetX implements a noise injection based on Differential Privacy (DP) with Gaussian noise (s = 1.2) in the process of training the model to guarantee confidentiality of the personal data. This research entails two partner organizations, which are a hospital and an insurance company, in an actual VFL configuration. The model is trained on 10 communication rounds in this federated setup, and the local optimization of each client is followed by the global aggregation step. Hospitals and insurers can learn fraud patterns without sharing their data. The proposed FraudNetX is a hybrid architecture which is composed of Feedforward Neural Networks (FFNNs) and transformer encoders. An adaptive weighting strategy has been applied to handle category imbalance concern and enhance recall of a few categories which are hard to detect, especially in fraud involving minorities, balancing the recall performance. The framework also includes a decision model that uses hospital data and claim behavior to classify each claim as legitimate, under review, or fraudulent. The experimental evaluation on the real-world dataset demonstrates that FraudNetX enhances the accuracy and F1-score of fraud detection (accuracy = 99.91%, F1 = 99.94%, ROC-AUC = 0.98) but does not violate data privacy.

## 1. Introduction

A major security concern in healthcare and insurance field is fraudulent transactions, since they result in losses and weaken public trust. When the data are confidential, the traditional method of detecting fraud has a number of problems. Sharing the data across the network to a central place for evaluation will require additional money to be spent on infrastructure. The risk of data breach comes from bringing data together. This is why some people are concerned about their data being kept secure, protected, and legal. Due to the lack of connected data, it is hard to detect various types of fraud involving industries. It has been highlighted in recent studies that the current centralized and semi-federated healthcare fraud detection systems do not provide any quantifiable privacy assurances and cannot deal with the challenges of vertically partitioned data across autonomous organizations [1].

Federated learning (FL) is a new approach that has recently emerged to allow the distributed training of machine learning models. Basically, federated learning lets various organizations team up to train a model, but their data remain on their own private servers. Subsequently, the point above resolves privacy concerns and prevents data from being stored in one location. Because of the fraud, FL can be trusted as an alternative to collaborate without revealing any personal information. There are many variants of FL based on different practices of sharing data. Horizontal federated learning means that all participating organizations share similar traits but have different datasets. For instance, hospitals will look for the same features in a patient, but these patients may come from different groups. Vertical Federated Learning is created for situations where organizations are part of the same population, but each one is different. While a hospital holds medical data of a person, an insurance company has access to the person’s financial records. A hospital and an insurance company can make models together by protecting each other’s valuable data.

Many researchers emphasized the significance of securely partitioning data across entities like hospitals and insurers by ensuring that sensitive patient and financial records remain protected during joint model training [2,3,4]. VFL is well-suited for fraud detection in healthcare and insurance domains where various entities hold different parts of a shared dataset. For example, hospitals and insurance firms can work together to expedite claim processing without directly exchanging personal information. This secure collaboration allows the joint analysis of trends and the detection of fraud while preserving data confidentiality. Nonetheless, available VFL-based fraud detection systems do not focus on optimizing the detection accuracy, but they also do not analyze privacy–utility trades or scale of deployment. FraudNetX, conversely, incorporates a hybrid Feedforward Neural Network (FFNN) and transformer-based and adaptive weighted privacy architecture with adaptable weighting and Gaussian noise mechanisms and privacy to provide both accuracy and quantifiable confidentiality. FTL is used when both the features and the sample spaces differ, and knowledge must be transferred across the domains. In this respect, privacy is a guarantee that the individual-level patient or claims-related information cannot be implied, reconstructed, or revealed in the model training or communication. In order to measure privacy, this paper uses Differential Privacy (DP) using Gaussian noise injection, with a privacy budget (ε, δ) = (1.0, 1 × 10^5^). These parameters tradeoff confidentiality and model performance, with more protection being granted by smaller or larger noise variance. This is suitable for complex cross-industry collaborations.

In this work, VFL has been focused on as an ideal approach for fraud detection in healthcare and insurance through its ability to handle vertically partitioned datasets. Stored medical information is often detailed in hospitals, while the detailed claim information is found in insurance companies’ databases. They are not authorized to share data in its raw state due to strict privacy regulations. Training an accurate model for fraud detection can be performed by sending updates to the models alone. Dhiman et al., (2022) [2] explained the distributed aspect of VFL, which follows privacy rules and enables the secure detection of fraud. Making use of various data sources improves VFL’s ability to identify fraudulent activities better than a system that works on its own. It can help to detect fraud among patients and workers at the hospital and insurance company, without affecting the privacy of anyone. In addition, data security is ensured, since VFL information remains at the local place where it is handled. As the information is kept locally, anyone looking after it has stronger control and security. The Differential Privacy (DP) approach is used for better protection of people’s private information. DP makes sure that VFL does not reveal personal data through the shared update. Zhao et al., (2022) [5] demonstrated how DP’s noise injection prevents adversaries from reconstructing sensitive details. Further, the authors in [6] combined homomorphic encryption with DP to reinforce privacy in federated learning settings. The major contributions of the proposed FraudNetX are highlighted below:The FraudNetX is a privacy-aware healthcare insurance fraud detector based on Vertical Federated Learning (VFL) between hospitals and insurers. The model combines Differential Privacy with Gaussian noise (σ = 1.2) to obtain measurable privacy guarantees (ε = 1.0, δ = 1 × 10^−5^).A hybrid framework based on a Feedforward Neural Network (FFNN) and transformer encoder is used, on which adaptive class weighting is applied to bring a balance between recall and precision of minority cases of fraud.As indicated in the experimental assessment, FraudNetX has a high accuracy and F1-score of more than 99% and at the same time has high data privacy in the federated setting.

## 2. Related Work

Experts in Vertical Federated Learning have managed to design systems that allow different parties to safely collaborate in machine learning. Researchers have managed to achieve positive results by developing models that make machine learning safer and more efficient without accessing the original data [7,8,9]. Even as the main data remain hidden, VFL supports unique companies to communicate and team up. During this process, the important point is the privacy of all participants in the joint learning process. Out of all privacy-protecting tools, Differential Privacy, homomorphic encryption, and secure multi-party computation are the most well-known [4,6]. It strives to hide private information during the process of collaborative learning. In addition, the sector is currently testing ideas on how to keep accuracy and communication between the parts of the data when it is divided among entities [10]. Further, the authors in [11] analyzed how to improve the VFL system by going through the iterative process. Moreover, the study helps to fix issues related to data leaks, inference attacks, and the time-consuming nature of computing. This makes VFL a decentralized machine learning solution for sensitive domains [12].

### 2.1. Data Preparation and Security Measures

The literature highlights secure data partitioning across healthcare entities. Moshawrab et al. [1] emphasized isolating patient and insurance records across the institutions to comply with privacy regulations. Dhiman et al. [2] reinforced this observation by demonstrating that VFL is compatible with global data protection laws. The privacy-constrained preprocessing methods in [3] manage nulls and outliers while preserving privacy. Zhang and Kreuter [4] have detailed privacy-preserving feature selection and normalization methods. Zhao and Wang [5] discussed the application of Differential Privacy on vertically partitioned datasets to hide sensitive data during training and inference. How to apply Differential Privacy alongside homomorphic encryption to provide two layers of protection was discussed in [6].

### 2.2. Model Training and Aggregation

In VFL, local model training allows entities to keep data private while contributing to a joint model. Anees et al. [7] demonstrated encrypted parameter sharing between parties. The Alternating Direction Method of Multipliers (ADMM) in [8] reduced communication loads by transmitting partial updates. Joshi et al. [9] provided an overview of global model aggregation mechanisms that ensure privacy during parameter fusion. Upreti et al. [10] focused on privacy-preserving aggregation steps for collaborative training. Zhao et al. [11] highlighted the effectiveness of Federated Averaging in maintaining accuracy in healthcare settings with non-uniform data. Hilberger et al. [12] optimized aggregation by enabling component-based dynamic model exchange and increased privacy. The Trusted Execution Environments (TEEs) implemented in [13] securely isolate sensitive computations for minimizing risks in high-stakes healthcare settings. Similarly, a threshold-based aggregation framework by Xu et al. [14] minimizes information leakage by enforcing cryptographic thresholds for updating contributions.

### 2.3. Iterative Improvement in Fraud Detection Applications

Iterative learning refines federated models across multiple update rounds. Chang and Zhu [15] illustrated how iterative strategies improve accuracy while minimizing data exposure. The FedAvg aggregator with Differential Privacy learning was used in [16] to optimize the results. Sánchez Sánchez et al. [17] evaluated the robustness of FL in non-IID data conditions, which are crucial for healthcare and insurance fraud detection. Matloob and Rahman [18] leveraged sequence mining within VFL for multi-institutional fraud detection. Zhou et al. [19] extended this approach by introducing FraudAuditor, a federated visual analytics system for real-time fraud monitoring. Kapadiya and Patel [20] combined blockchain with federated learning for tamper-resistant fraud systems. Du et al. [21] used a KNN-type estimation method, which preserves the privacy of incomplete medical records. Matloob et al. [22] continued this trajectory with sequence mining architectures. Li et al. [23] applied blockchain to cross-silo FL, and Gong et al. [24] introduced a multi-modal VFL framework using homomorphic encryption. Ghimire and Rawat [25] explored federated learning for cybersecurity applications in healthcare surveillance systems. Further, federated learning for cybersecurity applications in healthcare surveillance systems was described in [26].

Despite these advances, challenges such as computational overhead and multi-round protocol complexity remain. The proposed approach addresses these by combining VFL with Gaussian-based Differential Privacy, which remains computationally efficient. FraudNetX, a deep learning model, has been introduced for healthcare fraud detection. It includes techniques such as class weight balancing, sampling for minority fraud cases, and model ensembling to boost accuracy while preventing overfitting.

## 3. Methodology

The proposed framework enables privacy-preserving fraud detection in healthcare insurance using Vertical Federated Learning (VFL), where hospitals and insurance companies independently train models without sharing raw data. The Differential Privacy (DP) technique is applied to model gradients in order to ensure data confidentiality throughout the process. The Federated Averaging algorithm (FedAvg) aggregates local model updates to construct a global fraud detection model that classifies claims while preserving privacy. Figure 1 illustrates the architecture diagram of the proposed Vertical Federated Learning-based fraud detection system.

Figure 1 illustrates the safe information transmission and encrypted model update between the hospital and insurer nodes regarding the Vertical Federated Learning (VFL) arrangement.

### 3.1. Data Acquisition

The dataset used in this study is the publicly available Healthcare Provider Fraud Detection Analysis dataset from Kaggle. It includes healthcare provider interactions with insurance companies, focusing on potential fraudulent claims. Features include the following:Patient demographics;Provider details;Diagnosis codes;Procedure details;Reimbursement amounts.

The records in the dataset are classified into the following:Inpatient claims: Hospital-admitted patients;Outpatient claims: Routine checkups and diagnostics.

The model is evaluated using the file “Test_Outpatientdata-154296924375.csv”, which includes critical features like operation codes, patient age, and diagnostic data key indicators for uncovering fraud patterns. All Personally Identifiable Information (PII) is anonymized or removed to comply with the ethical standards and HIPAA regulations. Preprocessing also addresses missing values and class imbalance, which are critical for accurate fraud detection.

### 3.2. Data Partition

Vertical Federated Learning (VFL) has been employed to ensure that the sensitive medical and financial data from hospitals and insurance companies remain confidential while still supporting collaborative training. In VFL, each party owns a different set of features (e.g., Hospital Dataset and Insurance Dataset) for the same set of individuals.

#### 3.2.1. Hospital Dataset (Medical Data)

This dataset focuses on clinical and procedural details:BeneID, ClaimID;ClaimStartDt, ClaimEndDt (hospital stay duration);Provider, AttendingPhysician, OperatingPhysician;Diagnosis Codes (ClmDiagnosisCode_1–10);Procedure Codes (ClmProcedureCode_1–6);StayDuration, UniqueDiagnosisCount, UniqueProcedureCount;risc_score: A custom-engineered feature estimates claim risk based on medical complexity. The risc_score plays a vital role in detecting costly treatments that could indicate fraud or upcoding.

#### 3.2.2. Insurance Dataset (Financial Data)

This dataset contains financial metadata associated with claims:BeneID, ClaimID;IPAnnualReimbursementAmt, OPAnnualReimbursementAmt;IPAnnualDeductibleAmt, OPAnnualDeductibleAmt;Provider;claim_amount: A custom feature that reflects total reimbursements.

The insurance dataset focuses solely on financial aspects crucial for identifying inflated or suspicious claims.

#### 3.2.3. Rationale for Partitioning

Partitioning data into medical (hospital) and financial (insurance) segments respects the domain-specific sensitivity of each domain. Medical data remain confined within the hospital, while the financial records stay with the insurance provider. This setup aligns with the VFL architecture, which supports independent preprocessing and training. By exchanging only the encrypted model gradients, VFL enables both parties to retain full control over their respective datasets while still contributing to the shared fraud detection model. This architecture ensures the following:Privacy-preserving learning;Compliance with data governance policies;Effective cross-institutional fraud detection.

### 3.3. Data Preprocessing

#### 3.3.1. Hospital Dataset Preprocessing

The hospital dataset includes features such as patient stay duration, the number of unique diagnoses, and a risc_score estimating hospitalization risk.

Handling Missing Data: Rows with missing critical fields (e.g., risc_score) are excluded to prevent model performance degradation.

Feature Scaling: Continuous variables such as UniqueDiagnosisCount and StayDuration are scaled to maintain uniform influence across the features.

The RISC Score Range with its category are given in Table 1. Further the Risk Categorization is performed using Quartiles:Low Risk: ≤25th percentile;Moderate Risk: 25th–75th percentiles;High Risk: >75th percentile.

Target Variable Transformation: The risc_score categories are numerically encoded as 0 (low), 1 (moderate), and 2 (high) for classification compatibility.

Data Splitting: An 80:20 split is applied for training and testing.

This structure ensures robust training and generalization while maintaining balance across risc_score categories.

#### 3.3.2. Insurance Dataset Preprocessing

The insurance dataset focuses on financial information relevant to fraud detection.

Handling Missing Data: Rows with missing claim_amount or other critical values are removed.

Feature Scaling: Reimbursement fields (IPAnnualReimbursementAmt and OPAnnualReimbursementAmt) are standardized.

The Claim Amount Range with its category is given in Table 2. Further the Claim Amount Categorization is performed using Quartiles:Low Claim: <25th percentile;Medium Claim: 25th–75th percentiles;High Claim: >75th percentile.

#### 3.3.3. Integration and DataLoader Preparation

After preprocessing, hospital and insurance datasets are merged and structured for federated learning. Using PyTorch v2.7.0 DataLoaders, the system efficiently handles batching, shuffling, and parallel loading. These pipelines, covering feature scaling, categorical encoding, and clean test/train splits, ensure that the models are trained on high-quality, representative data. In a federated setting, this improves fraud trend identification while preserving data locality and security.

### 3.4. Feature Engineering

Feature engineering plays a crucial role in transforming raw healthcare and financial data into meaningful predictors of fraud. In a VFL setup, it also supports privacy-conscious intelligence.

#### 3.4.1. Claim Amount Calculation (Hospital Dataset)

The claim amount is computed to quantify financial risk:(1)$$\mathrm{Claim\_amount}=\left(R_{IP}+R_{OP}\right)-(D_{IP}+D_{OP})$$
where

$R_{IP},R_{OP}$ = Inpatient/Outpatient reimbursements;

$D_{IP},D_{OP}$ = Deductibles for inpatient/outpatient care.

Equation (1) defines the claim amount calculation combining inpatient/outpatient reimbursements.

#### 3.4.2. RISC—Risk Indicator Score Calculation (Insurance Dataset)

This derived feature estimates the financial exposure of each claim. A higher claim_amount may indicate potential fraud or upcoding. Equation (2) defines the risc_score, which measures patient-level medical complexity based on diagnosis and procedural diversity:(2)risc_score=α∗UniqueDiagnosisCount+β∗UniqueProcedureCount+γ∗StayDuration
where α, β, and γ are tunable coefficients optimized during model training. These mathematical equations back up the quantitative foundation for claim_amount and risc_score. The grid search was used to empirically optimize the coefficients (α, β, γ) on validation data in order to balance the impact of procedural, temporal, and diagnostic features. This metric reflects patient case complexity and potential misuse of resources, which are common indicators of fraud.

#### 3.4.3. Strategic Influence of Feature Engineering in VFL

These engineered features claim_amount and risc_score provide meaningful, privacy-safe insights into patient and claim behavior. They empower FraudNetX to detect anomalies across organizations without requiring direct data exchange. This approach creates a new standard for AI-enabled fraud detection: high precision, strong privacy compliance, and seamless cross-entity collaboration.

### 3.5. Differential Privacy

Following the preprocessing steps—such as handling missing values, encoding categorical features, and applying feature engineering (e.g., risc_score for hospitals and claim_amount for insurance)—privacy-preserving noise is applied to the data to prevent sensitive information leakage before model training. This noise masks individual-level information while retaining statistical properties necessary for fraud detection.

#### 3.5.1. Data Privacy Post-Preprocessing

In this scenario, Differential Privacy (DP) is used to protect against the leakage of sensitive data during model training. Noise is applied to the data to obscure individual-level records while preserving overall data patterns relevant for fraud classification. In federated learning, model gradients are sensitive and cannot be freely transmitted. To counter this, Gaussian noise is added to the computed gradients during the Differentially Private Stochastic Gradient Descent (DP-SGD) process.g’ = g + N(0, σ^2^C^2^)(3)
where g is the original gradient, σ is the noise multiplier, and C is the clipping norm. This makes it impossible for updates to disclose details about specific records. (ε, δ)-Differential Privacy is achieved by calibrating the noise magnitude. The maximum permitted privacy leakage probability is quantified by ε and δ, while σ denotes the Gaussian noise scale that establishes privacy strength. Stronger privacy (lower ε) is achieved with larger σ, but accuracy may suffer.

The privacy guarantee is implemented through Gaussian noise addition following the Differentially Private Stochastic Gradient Descent (DP-SGD) mechanism.

Hospital Dataset: Noise is applied to the risc_score feature to prevent reverse engineering of patient-level health information, while preserving model performance.

Insurance Dataset: Similarly, noise is added to the claim_amount to obscure financial data while maintaining fraud detection accuracy.

#### 3.5.2. Training Model with Privacy-Protected Data

Training of a model is performed on locally secured data following privacy-preserving preprocessing. The gradient updates are private with the help of the introduction of Gaussian noises and are aggregated safely among the clients so that no single contribution can be recovered. The following Algorithm 1 outlines the Differentially Private Federated Averaging (DP-FedAvg) procedure used in FraudNetX. Figure 2 shows noise perturbation visualization corresponding to this algorithm.
**Algorithm 1**: DP-FedAvgInput: K clients, global model w_0_, learning rate η, noise scale σ, clipping norm CFor each round t = 1...T:Server broadcasts $w_{t}$ to all clients        Each client i:                Computes gradients $g_{i}$ on local data                Clips gradients: $g_{i}=\nabla L\left(w_{i};D_{i}\right)$
                Adds noise: $g_{i}\leftarrow \frac{g_{i}}{max(1,\frac{{|\left|g_{i}|\right|}_{2}}{C})}$,                       ${g_{i}}^{\prime }=g_{i}+N(0,\sigma^{2}C^{2}I)$
where σ and C are as defined in S2.2 and S4                Updates weights: $w_{i}=w_{i}-\eta {g_{i}}^{\prime }$
Server aggregates: $w_{i+1}=\sum_{i=1}^{K}\frac{n_{i}}{n}w_{i}$
Output: Privacy-preserving global model $w_{T}$


Figure 2 illustrates the introduction of Gaussian noise in the DP-SGD to maintain privacy in the federated process. The Gaussian noise distribution used during the DP-SGD process is depicted in Figure 2. Individual-level data inference is prevented by the noise, which guarantees that gradient updates from every client remain indistinguishable. In this study, σ = 1.2, and the bell curve shows the additional Gaussian noise N(0, σ^2^C^2^).

#### 3.5.3. Privacy and Accuracy Balance

The application of Differential Privacy ensures that the training data are sufficiently obfuscated to avoid data leakage. By introducing privacy measures before training (rather than during), the system strikes a balance between data confidentiality and model accuracy. Hospital Dataset, the fraud detection model trained on risc_score, remains reliable while protecting individual patient data. Insurance Dataset, the model using claim_amount, continues to detect fraudulent claims effectively while maintaining financial privacy. This strategy ensures that the private data remains secure without compromising the integrity and effectiveness of the final fraud detection model. Empirical validation of the privacy–utility tradeoff (ε = 1.0, δ = 10^−5^, σ = 1.2). In Gaussian Differential Privacy, σ determines the tradeoff between privacy and utility. The total privacy loss ε over T rounds is computed via the composition theorem:ε_total = √(2T log(1/δ))/σ(4)

From the experimental results, ε = 1.0, δ = 10^−5^, and σ = 1.2 achieved an optimal privacy–utility balance. Higher σ improves privacy but slightly reduces accuracy.

The composition theorem was applied with T = 10 epochs and ε = 1.0 to derive the privacy guarantee and C = 1.0 and σ = 1.2 to obtain the privacy guarantee. In this design, the larger the σ, the more privacy increases, but the accuracy decreases slightly, and the larger the T, the greater the cumulative loss of privacy. The selected parameters hence guarantee an optimum privacy utility ratio in the FraudNetX framework.

### 3.6. Local Model Training

For local model training, the FraudNetX deep learning model is used to predict fraudulent claims in both the hospital and the insurance datasets. This architecture incorporates a Feedforward Neural Network (FFNN) and a transformer encoder to learn both non-sequential and sequential patterns from the data. Figure 3 illustrates the FraudNetX model architecture for fraud detection.

#### 3.6.1. Model Architecture

The architecture of FraudNetX consists of three major components:A Feedforward Neural Network (FFNN);A transformer encoder;A final output layer.

These three components enable the model to discover both static and temporal patterns relevant for fraud detection.

##### 3.6.1.1. Feedforward Neural Network (FFNN)

The FFNN processes input through fully connected layers to learn non-sequential relationships between features. Input data are passed through an input layer that maps feature vector X to a hidden representation of size hidden_size. The mathematical representation of the Feedforward Neural Network transformation is given ash = ReLU(Wx + b)(5)

##### 3.6.1.2. Transformer Encoder

The FFNN output is passed into the transformer encoder, which captures sequential relationships using multi-head attention. This mechanism enables the model to focus on multiple aspects of the input simultaneously. Two transformer encoder layers are stacked to enhance temporal pattern recognition. The final encoded representation is passed to a fully connected classification layer.

In the last stage of the encoder, a 1D feature is extracted, and it is passed to the final fully connected layer for classification.Output Layer—This module is a fully connected layer that takes the output from the transformer encoder and projects it to the final class labels.

The output layer maps this representation with three class labels: legitimate, review, and fraudulent. The transformer encoder utilizes multi-head attention defined as
(6)$$Attention\left(Q,K,V\right)=softmax\left(\frac{QKᵀ}{\sqrt{d_{k}}}\right)V$$
where Q, K, and V are query, key, and value matrices derived from the input embeddings.

The final classification output is computed as
(7)$$Z_{out}=W_{out}*A_{2}+b_{out}$$

#### 3.6.2. Training Procedure

Training is conducted across multiple epochs with the following steps:Zeroing gradients: Previous gradients are reset.Forward pass: Inputs are passed through the model to generate predictions.Loss calculation.

The Cross-Entropy Loss used in FraudNetX is calculated as(8)$$L=-\sum y_{i}*log\;(\hat{y_{i}})$$L Optimization follows Adam update rules with learning rate η=0.001(9)Precision=TP/(TP+FP)(10)Recall=TP/(TP+FN)(11)F1=2∗(Precision∗Recall)/(Precision+Recall)(12)Accuracy=(TP+TN)/(TP+TN+FP+FN)

To address class imbalance, class weights are applied in the loss function:

Class 0 (legitimate claims): Weight = 1.0;

Class 1 (fraudulent claims): Weight = 2.0;

Class 2 (review needed): Weight = 3.0.

Table 3 below shows the dimensional changes from the feature representational data to the final classification output in the FraudNetX architecture.

### 3.7. Federated Learning Process

This section describes the general federated learning process between hospitals and insurers, with both sides separately training local models and sharing encrypted gradients with each other to be summarized. Federated learning refers to the training of machine learning models among several distributed devices or server hardware (hospitals and insurance companies in this case) without sharing raw data. Every party teaches its local model from its own dataset. After training, model update gradients are delivered to a central server where Federated Averaging aggregates them to produce a global model. Then, the global model is sent back to the parties so that they can modify their local models. Iteratively repeating this technique allows collaboration while maintaining data privacy.

#### 3.7.1. Federated Averaging

On the basis of the federated process, this section describes how the local updates are pooled together using the Federated Averaging (FedAvg) algorithm to construct the global fraud detection model. Federated Averaging (FedAvg) has been applied to pool local models trained across distributed datasets. This preserves data privacy while collaboratively enhancing the global model. In Vertical Federated Learning (VFL), each client trains a local model and shares only model updates. These are then weighted and averaged based on the dataset size to update the global model. This method optimizes the accuracy of fraud detection while preserving privacy. FedAvg averages the parameters of the local model and then updates the global model. The training loop follows the DP-FedAvg scheme, combining local optimization, privacy-preserving gradient noise, and global aggregation. At the end of the training, the global model aggregates insights from all clients. Local data remain untouched to ensure privacy.

#### 3.7.2. Model Synchronization

It is based on the federated process, and the section explains how local updates are federated with each other through the Federated Averaging (FedAvg) algorithm to build the global fraud detection model. After federated training, model synchronization ensures that the global model integrates insights from both hospital and insurance clients.

The global FraudNetX model is initialized with

Input size = 2 (features: risc_score, claim_amount);Hidden size = 64;Output classes = 3 (legitimate, fraudulent, review needed).

The model is configured for inference on CPU (device = “cpu”). Both hospital and insurance datasets of independent evaluation are made ready to allow contemporaneous model evaluation. The global model is tested on the data of every client, and the sample measures the accuracy, the precision, the recall, and the F1-score to prove the generalization ability in different domains. Synchronization involves repeating FedAvg to aggregate client model weights into the updated global model. This integrates domain-specific learning with a unified, privacy-preserving representation. The resulting synchronized global model is cross-domain accurate and resistant to unseen claims. The model is recirculated to clients to constantly update them on the model in subsequent training rounds. Through synchronization, the model reflects shared intelligence from all participants and is capable of handling unseen claims with high precision.

### 3.8. Fraud Detection Model

This part consolidates the coordinated outputs of the hospital and insurance datasets and uses decision logic to understand that each claim is valid, needs to be reviewed, or is fraudulent. The global fraud detection model uses a structured decision dictionary that combines claim risk categories and reimbursement amounts to classify insurance claims as legitimate, review, or fraudulent. By integrating the hospital and the insurance data predictions, the system applies consistent logic to detect fraudulent activities.

#### 3.8.1. Decision Mapping for Fraud Detection

Mapping is defined based on the combinations of risk and claim categories, as shown Table 4:

The system takes risk_category and claim_category as inputs and returns the corresponding fraud status using decision mapping (Table 5). If a combination is missing (rare), the function returns “unknown”, indicating potential data issues.

#### 3.8.2. Mapping Predictions to Categories

**Hospital Model:** Predicts risk levels (0 = low_risk, 1 = moderate_risk, 2 = high_risk), which are then mapped with human-readable labels. The function apply_decision_logic() uses risk predictions and a default low_claim assumption to classify claims.

**Insurance Model:** Predicts claim levels (0 = low_claim, 1 = medium_claim, 2 = high_claim) and is similarly mapped with interpretable terms. Similarly, predictions use a default low_risk assumption, though this can be replaced with actual hospital risk categories if needed.

#### 3.8.3. Aggregating Results Across Datasets

The results of the prediction of fraud (legitimate, review, fraudulent) are counted on the hospital and insurance datasets using the Counter module of Python 3.12.5. The findings are subsequently integrated to give a summary of fraud. These comparative outcomes are visualized in domains in Figure 4 and Figure 5.

The graph illustrates the balance of contributions across domains by contrasting the fraud classification results between hospital and insurance models. The stacked bar chart shown in Figure 4 compares classification outcomes across hospital and insurance models and shows how each domain contributes to fraud detection. A logarithmic-scale distribution shown in Figure 5 highlights the dominance of “Legitimate” decisions, allowing easier visibility of fewer but critical “Fraudulent” and “Review” cases. The frequency of each fraud category is shown in this logarithmic plot, which also highlights infrequent but significant fraudulent and review cases.

#### 3.8.4. Performance Evaluation Metrics

The proposed model performance was evaluated using standard classification metrics (precision, recall, F1-score, support, and accuracy) consistent with sklearn’s metric definitions. Additional performance metrics used include ROC–AUC and precision–recall (PR) curves.True Positive Rate (TPR) = TP/(TP + FN)False Positive Rate (FPR) = FP/(FP + TN)

AUC represents the integral of TPR over FPR, summarizing the model’s ability to distinguish fraud from legitimate claims. The confusion matrix and PR plots of Hospital model are shown in Figure 6 and Figure 7, with numerical summaries tabulated in Table 6.

The model’s performance is evaluated using standard classification metrics derived from the sklearn.metrics module. Metric computation uses precision_recall_fscore_support() and accuracy_score() functions to evaluate each class—legitimate, review, and fraudulent—individually.

The computational structure of these metrics is consistent with precision_recall_fscore_support and accuracy_score functions from sklearn.metrics. Here, TP, TN, FP, and FN denote true positives, true negatives, false positives, and false negatives, respectively. A thorough evaluation of the model’s performance in the hospital and insurance domains is provided by the calculated results, which show metrics across the three fraud detection categories: legitimate, review, and fraudulent.

Accurate claim classification and improved interpretability of fraud detection results are made possible by the integrated decision logic system, which combines predictions from hospital and insurer models using claim and risk categories. The resulting metrics quantify the model’s capability to distinguish between legitimate, review, and fraudulent claims across both domains. Because of its broad healthcare provider–insurer claim structure, which is in perfect alignment with the Vertical Federated Learning (VFL) setup and represents hospitals and insurance entities as separate clients, the Kaggle Healthcare Provider Fraud Detection Analysis dataset was chosen for this study. Initially, alternative datasets like CMS Medicare and MIMIC-III were investigated; however, these either had access restrictions or lacked cross-domain linkages. Although the selected dataset has limited temporal information and the potential for synthetic data generation, it provides a sufficient diversity of claim types and fraud patterns. To improve model generalization, multi-institutional real-world claim datasets with validated fraud annotations may be used in future research.

## 4. Results

The evaluation results of the fraud detection model are presented through the analysis of both hospital and insurance datasets. The experiments are performed with an 80:20 train–test split in order to guarantee an unbiased test. The tuning of hyperparameters was performed solely on the training data, and there was neither an overlap nor a case of data leakage between the training and test sets. The model evaluation uses precision, recall, F1-score, support, and accuracy to measure its performance. The information about source code and data are provided in “Appendix A” section.

### 4.1. Hospital Dataset Model Evaluation

Figure 6 illustrates the confusion matrix of the hospital model, showing strong diagonal dominance that confirms reliable classification of hospital claims.

### 4.2. Classification Report for Hospital Model

**Hospital Dataset:** The model demonstrated superior performance when classifying genuine claims since it achieved a precision value of 0.988780 and a perfect recall score of 1.0. The model demonstrated strong precision (0.976190) together with a recall (0.823293) for detecting fraudulent claims. The model performed less accurately on review claims because it achieved a recall rate of 0.568704. The performance metrics of the hospital model appear in the Table 7, which demonstrates effective results together with identified weak points for enhancement.

The class-wise precision and recall are summarized in Figure 7. While recall needs to be improved for review cases, the hospital model shows nearly perfect detection for both fraudulent and legitimate claims. This visualization summarizes precision, recall, and F1-score values across hospital claim categories.

**Figure 7 sensors-25-07354-f007:**
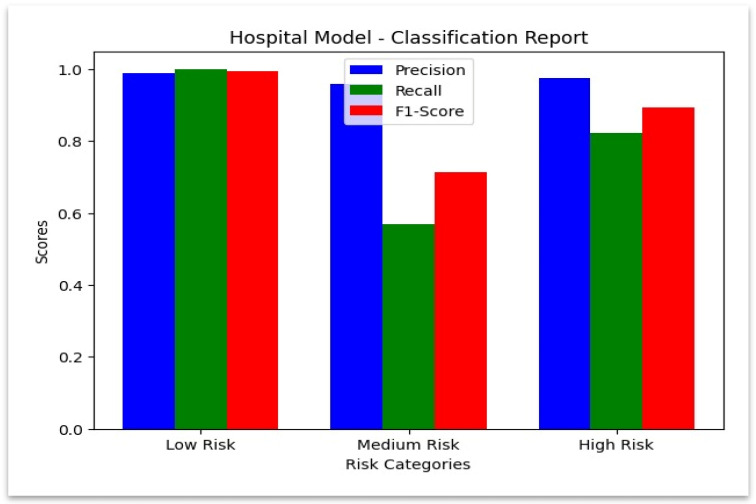
Classification Report—Hospital Model.

### 4.3. Hospital Model: Predicted vs. Actual Values

The model’s estimation accuracy is validated in Figure 8, which displays a close alignment between the predicted and true values. These findings create a privacy-preserving baseline for the insurance evaluation and validate the model’s dependability on hospital data.

### 4.4. Insurance Dataset Model Evaluation

The insurance model’s confusion matrix is displayed in Figure 9, emphasizing the precise identification of valid claims and the low rate of misclassification.

### 4.5. Classification Report for Insurance Model

**Insurance Dataset:** Similarly to the hospital dataset, the model has performed extremely well in predicting legitimate claims by achieving almost perfect precision (0.999690) and recall (0.998304). Fraudulent claims have a very high recall (1.0), which indicates that the model has identified all fraudulent claims in the insurance dataset. Additional ROC–AUC and precision–recall (PR) curves were calculated to evaluate model discrimination under class imbalance. Excellent separability of genuine and fraudulent claims was confirmed by both curves’ AUC values exceeding 0.99.

However, class imbalance caused the precision for fraudulent claims (0.548387) to drop, as the model slightly over-flags legitimate claims to make sure no fraud case is overlooked. The review category also shows a high recall (0.918033) with an F1-score of 0.879581, by demonstrating that the model could effectively identify claims that need review. Table 8 below highlights these performance metrics by illustrating the strengths of the model in fraud detection and areas where its precision could be refined further.

The class-wise performance for insurance data is summarized in Figure 10, which confirms high recall in every category.

### 4.6. Insurance Model: Predicted vs. Actual Values

Figure 11 demonstrates close alignment between predicted and actual values, indicating strong predictive stability.

**Accuracy:** The accuracy across both datasets reached high levels, as the model achieved 0.988279 accuracy for hospital claims and 0.99751 accuracy for insurance claims.

### 4.7. Accuracy–Privacy Tradeoff in FraudNetX

The tradeoff analysis uses the Differential Privacy parameters ε = 1.0 and δ = 1 × 10^−5^ with a Gaussian noise scale σ = 1.2 to quantify privacy. The tradeoff in privacy-protecting machine learning is that it inevitably erodes both the model’s accuracy and the confidentiality of data. Empirical testing has found that the higher the privacy level, the worse the model performance, as shown in Figure 12.

**No Privacy (Baseline):** Specifically, the model can achieve the best (99.9%) training accuracy without regard to privacy by using the full information in the dataset.

**Low Privacy:** Shows an incremental accuracy drop to 99.5%, suggesting a slight performance impact on the models.

**Medium Privacy:** The accuracy drops further to 99.0%, and it shows the influence of the change in model training with more noise.

For High Privacy, the model reaches 98.0% accuracy but witnesses the most relative performance drop due to enhanced privacy. The reported numbers make evident this fundamental tradeoff between privacy preservation and predictive ability in the context of federated learning settings. Stronger privacy constraints improve data confidentiality but also hurt fraud detection performance. Choosing a proper privacy level necessitates a tradeoff among compliance, fraud detection confidence, and system limitations. Moderate privacy setting (e.g., 99.0% accuracy) is found to be optimal as the best tradeoff between data security in combination with the capabilities of the model in identifying fraudulent transactions in a real-case scenario. These findings demonstrate that FraudNetX strikes a workable balance between model utility and confidentiality, maintaining over 99% accuracy while satisfying measurable privacy guarantees.

### 4.8. Performance Comparison of Healthcare Insurance Fraud Detection Models

The following comparative analysis, given in Table 9, shows the benchmarks of the proposed Hybrid FFNN + transformer model within a Vertical Federated Learning (VFL) framework against conventional machine learning and deep learning models.

#### 4.8.1. Analytical Findings and Observations

##### 4.8.1.1. Unparalleled Performance of the Hybrid FFNN + Transformer:

The proposed model achieves the highest accuracy (99.91%) and F1-score (99.94%), surpassing conventional ML and even sophisticated DL models. It demonstrates strong precision (99.92%) and recall (99.91%) by minimizing both false positives and false negatives [30].

##### 4.8.1.2. Limitations of Traditional ML Models:

Models like Logistic Regression, KNN, and Decision Trees struggle to understand complex fraud logic due to their inability to capture intricate relationships [27]. Their poor recalls render them ineffective in real-time fraud prevention scenarios [28].

##### 4.8.1.3. Deep Learning Advancements and Shortcomings:

Autoencoders and LightGBM improve fraud detection but still fall short in modeling nonlinear, contextual fraud patterns [29,30]. The hybrid model addresses these challenges by learning deep semantic dependencies within the transactional data. LightGBM and Autoencoders lessen reliance on manual features, but they are not interpretable and cannot ensure privacy in accordance with legal frameworks.

##### 4.8.1.4. LightGBM vs. Hybrid FFNN + Transformer:

The LightGBM framework performs well on structured data, but it fails to recognize subtle patterns in fraud. In contrast, the proposed architecture successfully learns multi-institutional fraud dependencies through transformer-based encoding (our work). Transformer-Driven Excellence in Fraud Detection: The transformer component captures long-range dependencies across hospital and insurance data by outperforming the traditional tree-based classifiers. This enhances generalization and recall and enables scalable and precise fraud detection (our work). Additionally, the transformer attention maps in the model contribute to Explainable AI (XAI) requirements by offering interpretable feature importance.

#### 4.8.2. Hybrid Performance Timeline: Advancements in Healthcare Fraud Detection

Figure 13 illustrates the timeline performance metrics (accuracy, precision, recall, F1-score) across the models, from early ML classifiers to the proposed VFL-based deep hybrid system.

##### 4.8.2.1. Performance Metric Visualization

This cohesive perspective aids in pinpointing turning points at which deep architectures start to surpass shallow learners (Figure 13).

X-axis: Fraud detection models in chronological order;Y-axis: Performance metrics—accuracy, precision, recall, and F1-score.

##### 4.8.2.2. Observed Trends

Traditional Models: Logistic Regression and KNN have poor recall and reduce real-world usability.Tree-Based Models: Decision Trees, XGBoost, and Random Forest show improved precision but struggle to generalize.Deep Learning: Autoencoders and LightGBM yield a better balance but miss hidden fraud relationships.Proposed Hybrid Model: Demonstrates superior recall and precision, leveraging transformer encoders within a VFL setup.

#### 4.8.3. Privacy-Preserving Collaboration via VFL

Beyond accuracy, this study makes a significant contribution by showing cross-entity collaboration that is prepared for compliance. Traditional centralized fraud detection is limited by data-sharing restrictions and regulatory constraints. Secure, cross-institutional training without data sharing is made possible by the VFL hybrid model, guaranteeing the following:Data privacy;Compliance with regulations (e.g., HIPAA);High detection accuracy.

FraudNetX can be practically deployed by integrating it into hospital–insurer infrastructures through federated nodes or secure APIs. Through differential privacy accounting, its modular design maintains auditability while facilitating scalability across healthcare networks.

## 5. Discussion

With 99.91% accuracy, 99.92% precision, and 99.91% recall, this study shows that the suggested hybrid FFNN + transformer architecture achieves state-of-the-art performance in detecting healthcare insurance fraud. This hybrid method outperforms both recent and conventional deep learning baselines, including LightGBM and Autoencoders. The model can learn both structured and contextual dependencies by incorporating a combination of transformer encoders and FFNN layers, which makes it possible to detect subtle fraud signatures that conventional algorithms frequently overlook. The suggested hybrid model maintains a balanced tradeoff between high precision and recall—minimizing both false positives and false negatives—whereas models like XGBoost and Random Forest achieve strong accuracy but limited recall. In the healthcare industry, where false alarms thwart valid claims and undetected fraud results in significant financial losses, this balance is essential.

## 6. Conclusions

The FFNN + transformer hybrid model proposed in this paper achieves sensible results on the task of detecting fraudulent Medicare claims. It is generally more effective than Logistic Regression, KNN, and Decision Trees, as well as Autoencoders and LightGBM. It illustrates how transformer systems can be adopted to look up details in structured medical and financial claims data. By handling false positives and false negatives, trust and traceability are maintained in fraud detection in real situations. By adding FFNN together with a transformer encoder, the model can perform well with the data of both hospitals and insurance companies. This model is suitable for fraud detection in a network of large hospitals. There are still two drawbacks in spite of these advantages. First, maximizing scalability and latency across dispersed nodes is necessary for real-time deployment. Second, in order for medical professionals to understand and have faith in AI-driven results, model transparency needs to be improved. Future research must focus on addressing these issues with Explainable AI (XAI) modules, such as attention-based interpretability or SHAP value analysis.

## 7. Limitations and Future Work

While the proposed FraudNetX framework achieved strong performance, some limitations still exist. Currently, the system has two partnering entities; it may be extended in the future to include more healthcare and insurance players to increase scalability. Overhead on communication and synchronization delay can affect real-time deployment and can be minimized by compressed update communication. Also, it would be beneficial to include the techniques of Explainable AI (XAI) like SHAP or attention-based interpretability to enhance the transparency and clinical acceptance of automated fraud detection. Future studies will investigate big, real-world data in order to test generalization within a wide range of healthcare settings.

## Figures and Tables

**Figure 1 sensors-25-07354-f001:**
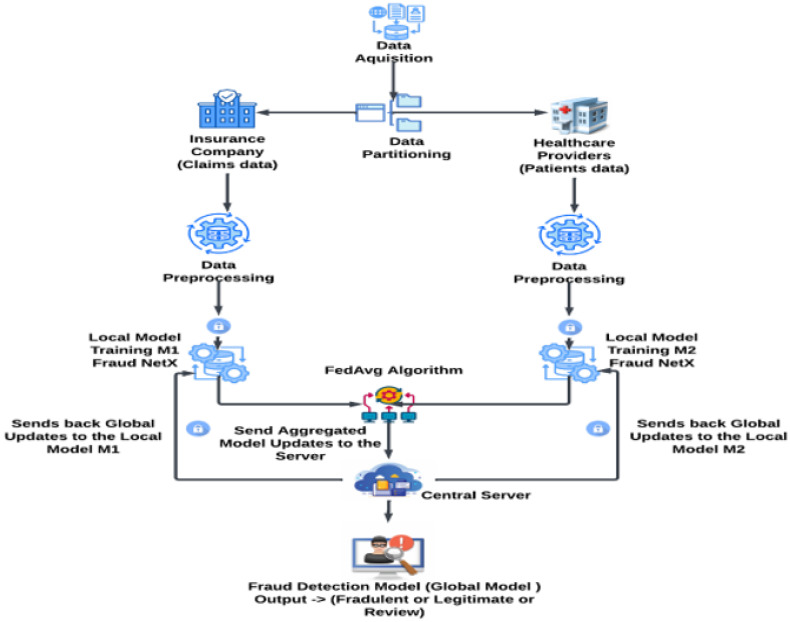
Architecture diagram of the proposed VFL-based fraud detection system.

**Figure 2 sensors-25-07354-f002:**
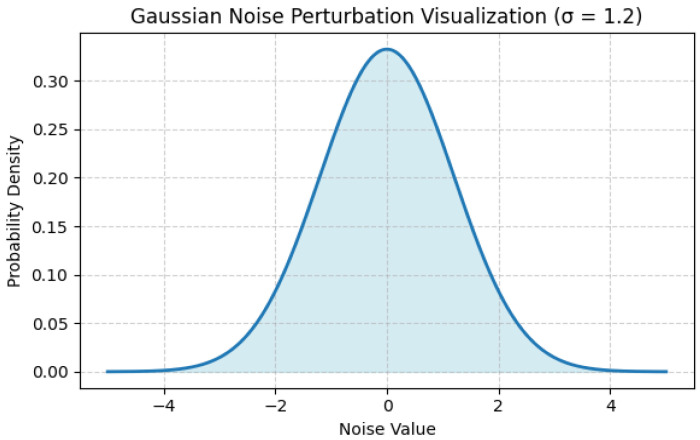
Visualization of Gaussian noise perturbation.

**Figure 3 sensors-25-07354-f003:**
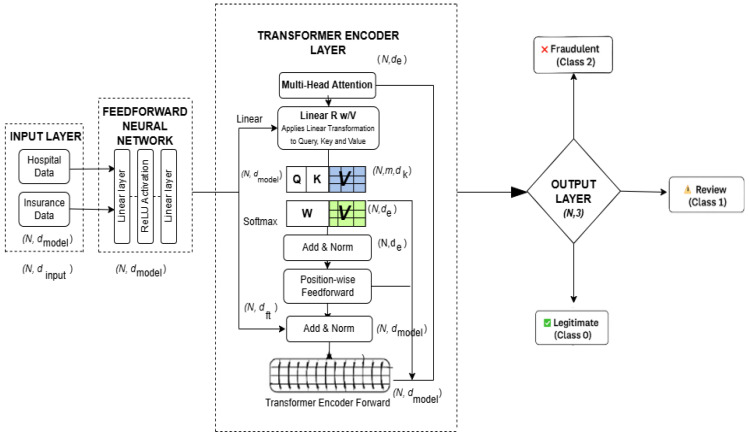
FraudNetX model architecture for fraud detection.

**Figure 4 sensors-25-07354-f004:**
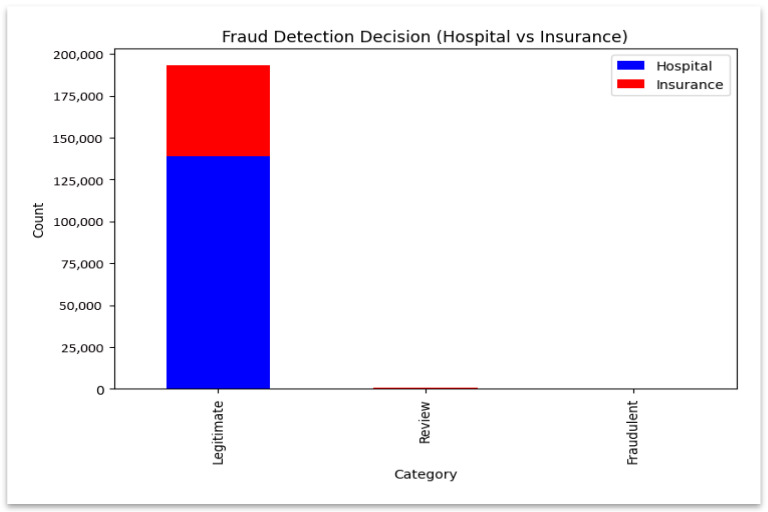
Fraud detection decision comparison.

**Figure 5 sensors-25-07354-f005:**
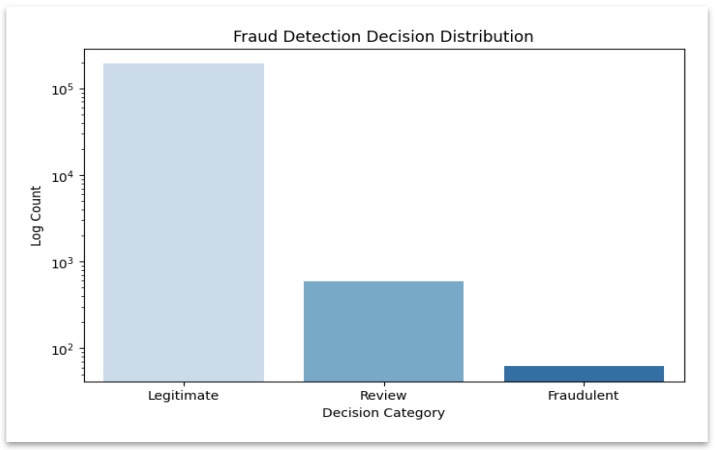
Fraud detection decision distribution.

**Figure 6 sensors-25-07354-f006:**
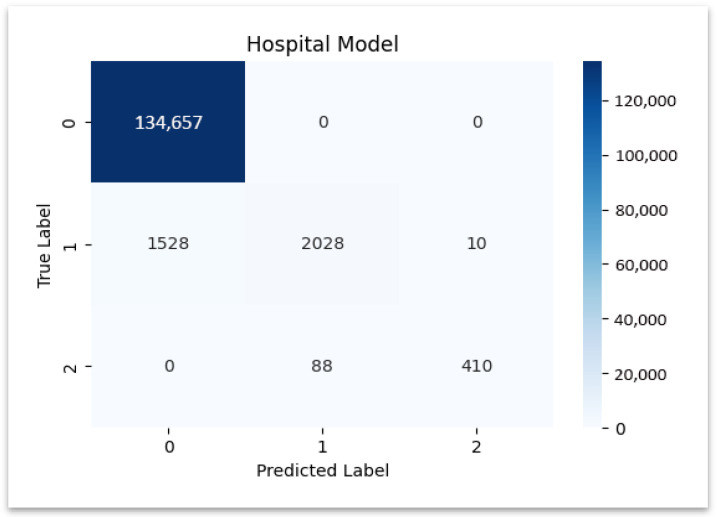
Confusion matrix—hospital model.

**Figure 8 sensors-25-07354-f008:**
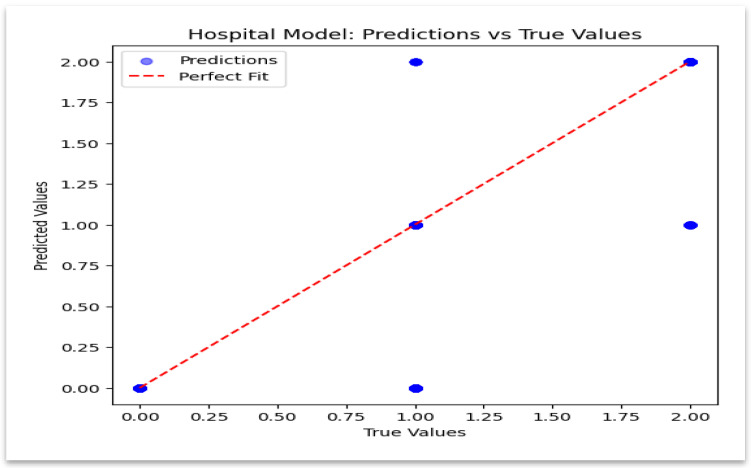
Hospital model—predicted vs. true values.

**Figure 9 sensors-25-07354-f009:**
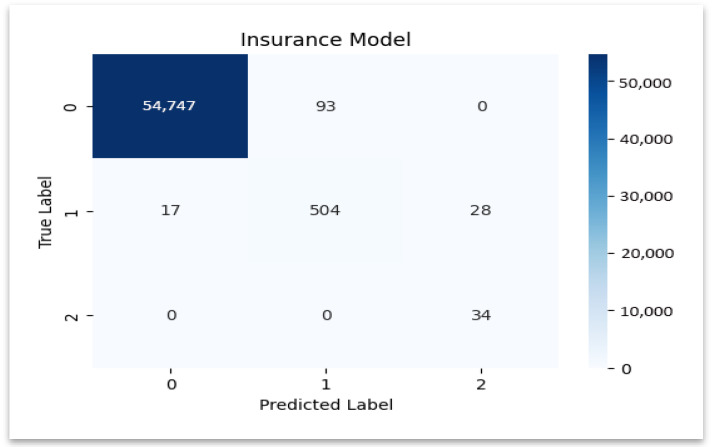
Confusion matrix—insurance model.

**Figure 10 sensors-25-07354-f010:**
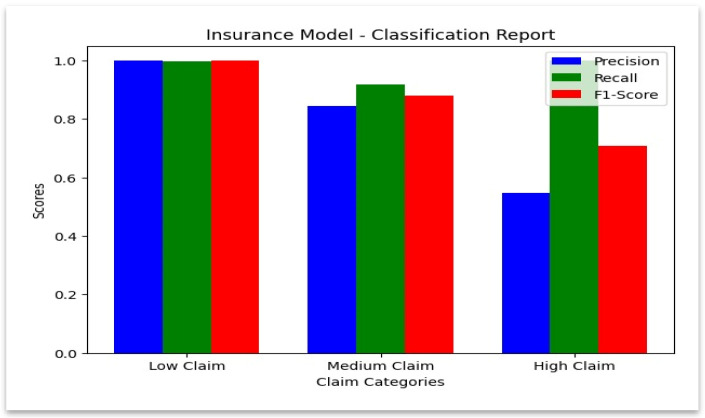
Classification report—insurance model.

**Figure 11 sensors-25-07354-f011:**
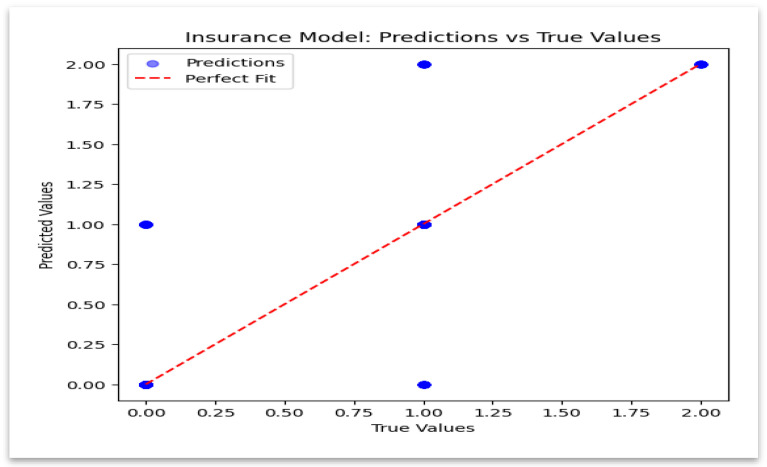
Insurance Model—Predicted vs. True Values.

**Figure 12 sensors-25-07354-f012:**
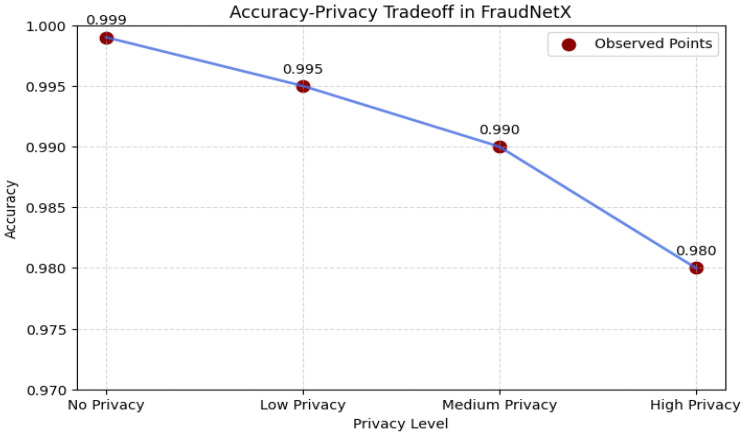
Accuracy–privacy tradeoff in FraudNetX model.

**Figure 13 sensors-25-07354-f013:**
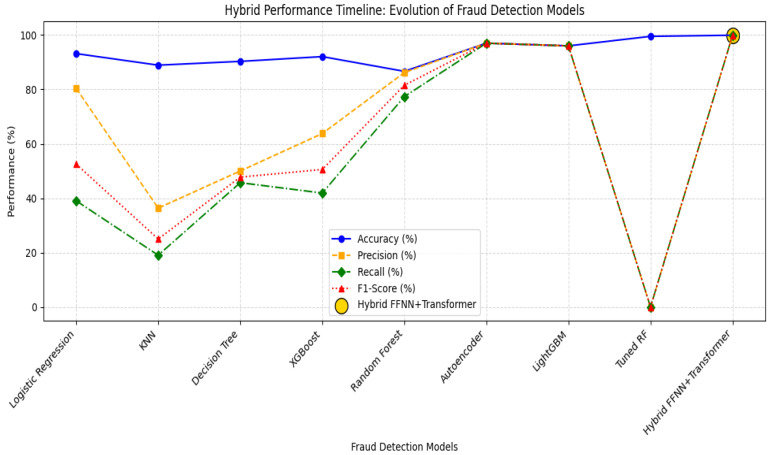
Hybrid performance timeline: evolution of fraud detection models.

**Table 1 sensors-25-07354-t001:** Categorization of RISC score.

RISC Score Range	Category
0.7 ≤ score ≤ 4.3	Low Risk
4.3 < score ≤ 7.9	Moderate Risk
7.9 < score ≤ 14.5	High Risk
Others	Unknown

**Table 2 sensors-25-07354-t002:** Categorization of claim amount.

Claim Amount Range	Category
1000.0 ≤ amount < 50,000.83	Low Claim
50,000.83 ≤ amount < 100,000.66	Medium Claim
100,000.66 ≤ amount ≤ 261,219.5	High Claim
Others	Unknown

Target Encoding: Categories are encoded as 0 (low), 1 (medium), and 2 (high). Data Splitting: The Same 80:20 split strategy has been applied for consistent evaluation.

**Table 3 sensors-25-07354-t003:** Dimensional representations in the FraudNetX model.

Component	Mathematical Notation	Description
Input Layer	$\left(N,\;d_{input}\right)$	The raw feature input from hospital and insurance data. NNN represents the batch size, and $d_{input}$ is the total number of features from both sources.
Feedforward Neural Network (FFNN) Output	$\left(N,\;d_{model}\right)$	The transformed feature representation after passing through the FFNN. $d_{model}$ is the latent dimension used for subsequent processing.
Query (Q), Key (K), Value (V) Matrices	$\left(N,m,d_{k}\right)$	The projected representations of the input features for the self-attention mechanism. Here, *m* is the number of attention heads, and $d_{k}$ is the dimension of each head.
Attention Weights (W) Before Softmax	$\left(N,\;d_{e}\right)$	The computed attention scores before applying softmax normalization. $d_{e}\;$represents the embedding dimension of the transformer encoder.
Softmax Attention Output	$\left(N,\;d_{e}\right)$	The final attention output after applying the softmax function for maintaining the embedding dimension $d_{e}$.
Add and Norm Layers	$\left(N,\;d_{e}\right)$	The residual connections are followed by layer normalization to ensure stable training.
Position-wise Feedforward Network Output	$\left(N,\;d_{model}\right)$	The output of the position-wise FFNN applied after self-attention, restoring the $d_{model}$ dimension.
Transformer Encoder Forward Output	$\left(N,\;d_{model}\right)$	The result encoded after several self-attention and feedforward transformations.
Final Classification Layer	(N,3)	The output array contains scores of the probability distribution over three classes: legitimate, review, and fraudulent.

**Table 4 sensors-25-07354-t004:** Risk and claim categories.

Risk Categories	Claim Categories
Low	Low
Moderate	Moderate
High	High

**Table 5 sensors-25-07354-t005:** Fraud detection mapping based on risk and claim categories.

Risk Category	Claim Category	Fraud Detection Outcome
Low	Low	Legitimate
Low	Moderate	Review
Low	High	Fraudulent
Moderate	Low	Legitimate
Moderate	Moderate	Review
Moderate	High	Fraudulent
High	Low	Legitimate
High	Moderate	Review
High	High	Fraudulent

**Table 6 sensors-25-07354-t006:** Metrics, formulas, and descriptions.

Metric	Formula	Description
Precision	$Precision=\frac{TP}{TP+FP}$	Fraction of relevant positive prediction
Recall	$Recall=\frac{TP}{TP+FN}$	Coverage of actual positives
F1-Score	$F1=2*\frac{Precision*Recall}{Precision+Recall}$	Harmonic mean of precision and recall
Support	$Support\left(c\right)=\sum_{i=1}^{N}1(y_{i}=c)$	Count of true instances per class
Accuracy	$Accuracy=\frac{\left(TP+TN\right)}{\left(TP+TN+FP+FN\right)}$	The overall proportion of correct predictions.

**Table 7 sensors-25-07354-t007:** Performance metrics of the hospital model.

Category	Hospital Model				
	Precision	Recall	F1-Score	Support	Accuracy
Legitimate	0.9887	1.0000	0.9943	134657	0.9882
Review	0.9584	0.5687	0.7138	3566	0.9882
Fraudulent	0.9761	0.8232	0.8932	498	0.9882

**Table 8 sensors-25-07354-t008:** Performance metrics of the Insurance model.

Category	Insurance Model				
	Precision	Recall	F1-Score	Support	Accuracy
Legitimate	0.9996	0.9983	0.9989	54840	0.9975
Review	0.8442	0.9180	0.8795	549	0.9975
Fraudulent	0.5483	1.0000	0.7083	34	0.9975

**Table 9 sensors-25-07354-t009:** Comparative evaluation of algorithms on healthcare insurance fraud detection benchmark.

Model	Accuracy (%)	Precision (%)	Recall (%)	F1-Score (%)	Key Insights
Logistic Regression [27]	93.16	80.39	39.05	52.56	Moderate accuracy; poor recall leads to high false negative rates.
K-Nearest Neighbors [27]	88.91	36.36	19.05	25.00	High misclassification; ineffective for fraud.
Decision Tree Classifier [27]	90.30	50.00	45.71	47.76	Improved recall; still lacks precision.
XGBoost (XGB) [27]	92.05	63.77	41.90	50.57	Better generalization, weak recall.
Random Forest (Depth = 25, Estimators = 1000) [28]	86.64	86.19	77.35	81.53	Balanced results; still not optimal.
Autoencoder [29]	97.00	97.00	97.00	97.00	Deep learning anomaly detection improves performance.
LightGBM [30]	96.00	96.00	96.00	96.00	Strong for structured data; struggles with complex patterns.
Tuned Random Forest [30]	99.50	-	-	-	Near-perfect accuracy; risk of overfitting.
Proposed Hybrid FFNN + Transformer in VFL	**99.91**	**99.92**	**99.91**	**99.94**	State-of-the-art accuracy; minimal false positives.

Note: The proposed model operates under Differential Privacy (ε = 1.0, δ = 1 × 10^−5^), ensuring comparable evaluation with non-private baselines.

## Data Availability

The data presented in this study are available on request from the corresponding author. The data are not publicly available due to privacy.

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
