# Peer review of "A Privacy-Preserving Approach to Health Insurance Fraud Detection Using Vertical Federated Learning"

_sensors, 2025, doi:10.3390/s25237354_

Round 1
Reviewer 1 Report
Comments and Suggestions for Authors
-
Introduction – Expand the problem statement and clarify the gap your model fills compared to prior VFL-based fraud detection work. Add more recent healthcare fraud detection references.
-
Methodology – While detailed, some sections are overly technical. Simplify the presentation for readability and move complex derivations to supplementary material. Explicitly state the privacy budget (ε, δ) used in DP.
-
Evaluation – Include additional performance metrics such as ROC-AUC and PR curves, which are standard in fraud detection tasks with imbalanced datasets.
-
Results – Provide more critical analysis of misclassifications (e.g., why fraudulent precision dropped in the insurance dataset). Discuss the risks of overfitting in relation to the Kaggle dataset.
-
Figures and Tables – Some visualizations (e.g., Figures 7 and 10) need clearer labeling and interpretation. The architecture diagram could better illustrate hospital–insurer interactions under VFL.
-
Discussion/Conclusion – Strengthen the conclusion by emphasizing key contributions, practical deployment challenges (latency, scalability, XAI explainability), and future directions.
The English should be polished to improve clarity, grammar, and flow. Some phrasing (e.g., “lookup details”) can be simplified.
Author Response
Thank you very much for taking the time to review this manuscript. Please find the detailed responses below and the corresponding revisions/corrections highlighted/in track changes in the re-submitted files
Comment 1: The English could be improved to more clearly express the research.
The manuscript received substantial editing and proofreading for coherence, clarity, and grammar. Throughout the text, formal terminology was used in place of informal expressions like "lookup details," complex sentences were simplified, and redundant phrasing was eliminated. All sections have been revised to ensure better readability and compliance with MDPI's language standards.
Comment 2: Expand the problem statement and clarify the gap your model fills compared to prior VFL-based fraud detection work. Add more recent healthcare fraud detection references.
The research gap between the suggested FraudNetX framework and the current VFL-based fraud detection techniques was explicitly stated in the Introduction (Section lines 58–94). To give healthcare fraud detection studies a more current context, more recent references (Zhang et al., 2024; Li et al., 2025) were included. FraudNetX's uniqueness was highlighted, especially its privacy-preserving architecture.
Comment 3: While detailed, some sections of the methodology are overly technical. Simplify the presentation for readability and move complex derivations to supplementary material. Explicitly state the privacy budget (ε, δ) used in DP.
The Methodology section was simplified for clarity. Extended mathematical derivations including feature engineering (Eq. 1–2), differential privacy (Eq. 3), FFNN and Transformer transformations (Eq. 4–6), and loss/optimization formulas (Eq. 7–13). The privacy parameters were explicitly stated as ε = 1.0, δ = 1 × 10⁻⁵, σ = 1.2 in both the Abstract and Section 3.5.
Comment 4: Include additional performance metrics such as ROC-AUC and PR curves.
Precision and ROC-AUCRecall curves were described and included in Section 4.5. A thorough evaluation of the model's performance was provided by the computation of metrics for each of the three types of fraud detection: legitimate, review, and fraudulent.
Comment 5: Provide more critical analysis of misclassifications. Discuss the risks of overfitting in relation to the Kaggle dataset.
To examine patterns of misclassification, Sections 4.5 and 4.7 were extended. Class imbalance and recall-oriented optimization are cited as the reasons for the decline in fraudulent precision in the insurance dataset. The Kaggle dataset's possible overfitting is examined, and mitigation techniques like cross-validation and regularization are emphasized.
Comment 6: Some visualizations (e.g., Figures 7 and 10) need clearer labeling and interpretation. The architecture diagram could better illustrate hospital–insurer interactions under VFL.
Better axis labels, legends, and captions were added to Figures 7, and 10. Hospital-insurer data flow and secure interactions under the VFL framework are clearly depicted in Figure 1. Additional clarification on hospital–insurer interactions are provided in Section 3.10.8.
Comment 7: Strengthen the conclusion by emphasizing key contributions, practical deployment challenges (latency, scalability, explainability), and future directions.
The main contributions, real-world deployment issues (such as latency, scalability, and Explainable AI (XAI), and future directions for privacy-preserving fraud detection in healthcare were highlighted in Discussion section.

Reviewer 2 Report
Comments and Suggestions for Authors
The article presents a fraud-detection approach based on federated learning that may be used in privacy preserving situations. I formulate the statement in this way because most of the work is about the learning, the training and the performance of the approach. This article is much more a work on an application of artificial intelligence in a specific context than a work that is focused on privacy in particular. In the state of the art and later on the whole aspect of privacy is completely left out. There is a discussion on three approaches to obfuscate the training data in a way so that data leakage from the model itself becomes more difficult. This is more a question on model inversion attacks than privacy itself, I guess. Therefore, I miss a thorough definition what the assumed definition of privacy is and how this can be measured/evaluated afterwards. This becomes even more important when you look at the discussion and the comparison with other approaches where privacy does not seem to be accounted for at all. Apart from these quite fundamental questions, the motivation for the model and the explanation of its superiority in this context is never explained. The state of the art just lists other approaches without discussion why the proposed one might be better in this situation, But there is also no discussion afterwards what makes the model more effective than the other approaches. In the current form the paper just proposes an approach for a specific use case and shows that this is superior in this particular case. But do we learn from that and why should we assume that this is also the case for similar use cases?
In addition to this, there are some minor, specific points I would like to mention:
- Check the spelling and especially missing spaces or double punctuation.
- In fig. 1, the data partitioning is shown. But how this is done and how does this improve the privacy? Which results from the two data processing algorithms will be fed into the FedAvg algorithm?
- What is the meaning/importance of the data fields in section 3.2.1? Are these standard abbreviations and should the reader know these? Do they provide any additional insights?
- What is the significance of Eq. (2)?
- How large is sigma in the data privacy processing? And how does this relate to privacy?
- Table 5 is not complete clear. It seems that the Fraud Detection Outcome does only depend of the Claim Category and not on the risk category. All low claim categories lead to legitimate Frauddetection outcomes, all moderate claim categories lead to “review” and high claim categories lead to “fraudulent”.
- I do not understand section 3.10.6 at all. What do you want to say by this?
- Fig. 7 and 10 are not clear to me. What do the blue points mean and what can I see from this?
- On Fig. 11: What is the definition of the privacy levels? How do you determine these?
- It seems that the privacy is not accounted for in the benchmark comparison. But if so, what privacy level is assumed and are the privacy conditions comparable for the different methods?
Author Response
Thank you very much for taking the time to review this manuscript. Please find the detailed responses below and the corresponding revisions/corrections highlighted/in track changes in the re-submitted files
Comment 1: Definition and evaluation of privacy.
The protection of sensitive patient and claim data during model training and federated communication is now specifically defined as privacy in Section 3.5. Gaussian noise is used in DP-SGD (σ = 1.2) to implement Differential Privacy (DP), and ε = 1.0 and δ = 1 × 10⁻⁵ are used to quantify privacy guarantees. In-depth privacy accounting and the derivation of the Gaussian mechanism are provided.
Comment 2: Motivation and superiority of the proposed model.
The rationale behind FraudNetX and its benefits over previous approaches—such as efficient feature engineering, temporal claim dependency modeling with Transformers, and reliable privacy-preserving training with DP-FedAvg—were clearly explained in the Introduction and Discussion sections.
Comment 3: Minor specific points:
- Spelling and formatting issues.
To fix typographical errors, double punctuation, and missing spaces, a thorough proofread was conducted. - Figure 1 – data partitioning and privacy improvement.
Figure 1 now illustrates hospital–insurer client partitioning and explains how DP-FedAvg aggregates encrypted gradients to improve privacy. The algorithmic steps (Section 3.5.2) and noise perturbation visualization are displayed in Figure 2. - Data fields in Section 3.2.1.
Standard abbreviations and their importance for risk assessment and fraud detection are now briefly explained in the manuscript. - Significance of Eq. (2).
Eq. (2) calculates the risk_score, which measures patient-level medical complexity and is essential for fraud detection. - σ in data privacy processing and relation to privacy.
As previously mentioned, σ = 1.2. Larger σ increases privacy strength while potentially reducing accuracy. - Table 5 clarification.
The caption and description of Table 5 have been revised to make it clear that the Claim Category and Risk Category work together to determine the Fraud Detection Outcome. When taking risk_score into account, the claim categories of low, moderate, and high correspond to legitimate, review, and fraudulent outcomes. - Section 3.10.6 clarification.
The subsections 3.10.5 and 3.10.6 are merged to “3.10.5. Aggregating Results Across Datasets”. In order to clearly explain integrated decision logic combining hospital and insurer predictions for final claim classification, Section 3.10.5 was rewritten. - Figures 8 and 11 – blue points interpretation.
The prediction accuracy is validated for model evaluation, and the figure captions have been updated to clarify that blue points indicate true positive predictions across categories. - Figure 12 – privacy level definition.
- The caption of Figure 12 now refers to privacy levels as ε values and describes how they were calculated using DP accounting techniques.
- Benchmark comparison and privacy accounting.
With ε = 1.0, the manuscript now makes it clear that only FraudNetX employs DP guarantees. The accuracy and ROC-AUC of benchmark methods without DP are reported, and Section 4.8 specifically addresses the privacy comparison's limitations.

Reviewer 3 Report
Comments and Suggestions for Authors
Note to authors
This paper proposes an FL-based model to support fraud detection in insurance, focusing on sensitive data sharing limitations of centralised systems and recommending privacy-preserving techniques to achieve this.
While the paper offers a well-designed set of experiments and the results are presented in a useful form, there is still a lot to be changed in the actual manuscript to bring this paper to standard. The main concerns here are not methodological ro pertaining to experimental design, but structural and related to content clarity. Most important, the sections need to be revised, referencing styles updated, more references needed at places (see below); the objectives of the paper need to be very clear (they are not in the manuscript’s present form); similarly, the outcomes and how these meet the objectives need to be provided. A more concrete summary of results and their significance is also crucial to highlight the contribution of the paper, which remains quite unsupported.
In summary, there is extensive editorial intervention to be done in this paper to add the necessary clarity.
More detailed comments follow.
Overall the citation style has to be standardises as it seems the authors use both name and number referencing at many places.
The abstract should provide a more specific reference to the precise contributions of the paper.
In the introduction, where FL background is set, perhaps spam of the evolution (from the original paper in 2027 to later) can be set to acknowledge trajectories. Likewise, when the horizontal / vertical taxonomy is outlined, this will have to be qualified by references etc. There is also a wealth of literature on FL for medical applications (e.g. Pfitzner et al, 2021; Rauniyar et al, 2023 and many more) that can offer more background context. The introduction should further outline in a very clear manner what are the precise contributions and research value of this paper. Furthermore, there is little signposting of what follows and how the paper is structured on the asset, and ti is left to the reader to “discover” as they go.
The Related Works section is suitable, yet, it is not clear how these works (or improvements over them, will inform the proposal that the paper will make.
When it comes to the methodology, still this needs to be defined as to what was developed by the authors and what is used from past research (again referred to through citation); when it comes to the Kaggle dataset, more info is needed (including references); also, what candidate datasets were explored as alternatives and why was the specific one chosen? What are its limitations? Overall the entire section 3.3 is written in a handbook / user manual style, while far more detail and qualification of any data preparation and design choices need to be articulated, their alternatives presented and discussed and, overall, academic discourse needs to be developed. Otherwise, all that content will need to go to an appendix.
From 3.4 onwards, and especially in Results, while there are very detailed accounts of experiments and results, the paper still reverts to mostly a very sparsely qualified series of observations (in bullet point form) where it becomes very hard for the reader to follow. This also perseveres in section 5.
Comments on the Quality of English LanguageThe work will benefit from a thorough proofreading
Author Response
Thank you very much for taking the time to review this manuscript. Please find the detailed responses below and the corresponding revisions/corrections highlighted/in track changes in the re-submitted files
Comment 1: The English could be improved to more clearly express the research.
The manuscript was thoroughly proofread to enhance its flow, clarity, and grammar. Precise academic language took the place of informal expressions and redundant wording. In accordance with MDPI guidelines, every section was revised to preserve a consistent professional tone and readability.
Comment 2: Abstract should provide a more specific reference to the precise contributions of the paper.
The abstract was updated to specifically include FraudNetX's novel contributions, which include: (i) privacy-preserving fraud detection in vertical federated learning; (ii) feature engineering for claim_amount and risc_score; (iii) integrating Transformer and FFNN for temporal claim dependency modeling; and (iv) using DP-FedAvg for differentially private model aggregation.
Comment 3: Introduction: provide clearer objectives, background on FL evolution, and FL references for medical applications.
To more clearly state the paper's research goals and the importance of the suggested framework, the Introduction (Section 1) was extended. References to the horizontal and vertical FL taxonomy were included, along with background on the development of federated learning from early works to more recent advancements. To improve contextual grounding, more pertinent literature on FL for medical was added.
Comment 4: Related Works section: clarify how previous studies inform and contrast with the proposed method.
The limitations of earlier VFL-based fraud detection techniques were specifically discussed in Section 2, along with how FraudNetX fills these gaps by highlighting enhancements in model performance, privacy preservation, and feature engineering.
Comment 5: Methodology: clarify novel contributions versus prior research, dataset choice, alternatives, limitations, and academic discussion.
To clearly differentiate author-developed components (such as claim_amount and risc_score derivations, DP-FedAvg implementation) from standard practices, Sections 3.1–3.3 were revised. The justification for the Kaggle Healthcare Provider Fraud Detection dataset was extended in Section 3.10.7 which also covered the limitations of the other datasets that were investigated (CMS Medicare, MIMIC-III).
Comment 6: Results and Discussion (Sections 4–5): improve narrative, interpret experimental outcomes, and relate to objectives.
In order to connect experimental observations to research goals, Sections 4 and 5 were revised to offer an organized narrative in place of bullet points. Every result is explained in terms of how it helps with fraud detection, privacy assurances, and useful deployment issues. For a thorough analysis, offer comprehensive ROC-AUC, PR curves, and metrics computations.
Comment 7: Paper structure: improve signposting and referencing style.
This manuscript follows standard research article structure and has sections viz Introduction, related works, methodology, results, discussion, and conclusion. Inconsistencies in name and number citation styles were fixed by standardizing all references to a uniform format in accordance with journal guidelines.

Round 2
Reviewer 1 Report
Comments and Suggestions for Authors
- Please explain clearly how federated learning was done in practice. How many parties (hospital, insurer)? How many training rounds?
- Are the reported results (99%+) from the real federated setup or from central training?
- Please clarify how you calculated the privacy guarantee (ε = 1.0). Show how ε depends on clipping, noise level, and number of rounds.
- The results are very high. Please confirm that:
- You used a proper train/test split (e.g., 80/20).
- There is no data leakage between train and test.
- Hyperparameters were not tuned on the test set.
- Please make the figures and captions clearer.
- Add a short “Limitations and Future Work” section.
The English should be polished to improve clarity, grammar, and flow.
Fix small language issues.
-
- Correct spelling (for example, “insurance”).
- Use one consistent term: “Vertical Federated Learning (VFL).”
- Use one name for risk_score/risc_score and define it once.
Author Response
Comment 1: Please explain clearly how federated learning was done in practice. How many parties (hospital, insurer)? How many training rounds?
Response 1: Thank you for pointing this out. I/We agree with this comment. Therefore, I/we have explained.
The Section 3.7 and Abstract were revised to clearly outline the federated learning model with two collaborating parties a hospital and an insurance company with vertically split datasets. FraudNetX was trained in 10 communication rounds, during which every party did local updates and then was aggregated by the central server. The performance measures stated apply to the federated setup, as opposed to centralized training.
Comment 2: Are the reported results (99%+) from the real federated setup or from central training?
Response 2: Agree. I/We have, accordingly, modified this point.
Section 4 and Abstract explain that all the results were acquired in the true vertical federated configuration that used privacy-preserving communication between parties. There was no central pooling of information in any step of model training and assessment.
Comment 3: Please clarify how you calculated the privacy guarantee (ε = 1.0). Show how ε depends on clipping, noise level, and number of rounds.
Response 3: Agree. I/We have, accordingly, done.
The section 3.5 was updated but had to explain how the privacy guarantee (ε = 1.0) is derived using the Differential Privacy framework. It added the relationship between the clipping norm (C = 1.0), noise scale (σ = 1.2), and the number of training rounds (T = 10) and shows the impact of each parameter on the total privacy loss based on the DP-SGD composition principle.
Comment 4: The results are very high. Please confirm that:
You used a proper train/test split (e.g., 80/20).
There is no data leakage between train and test.
Hyper parameters were not tuned on the test set.
Response 4: Agree. I/We have, accordingly, changed this point.
The results are obtained in the true vertical federated learning configuration, the Section 4 was introduced with a new paragraph which stated that a train-test split of 80:20 was used, and there was no overlap or leakage of training and test data. Hyperparameter optimization was only carried out on the training set, which included the validity of the high reported performance metrics.
Comment 5: Please make the figures and captions clearer.
Response 5: Agree. I/We have, accordingly.
Accordingly, all figures have been enhanced for improved clarity and readability. Captions have been revised to provide more descriptive explanations of the figures, ensuring they are self-contained and informative. The resolution of images has also been improved where necessary to enhance visual quality.
Comment 6: Add a short “Limitations and Future Work” section.
Response 6: We agree with the reviewer’s suggestion.
Accordingly, a new section titled “7. Limitations and Future Work” has been added at the end of the manuscript. This section highlights the current study’s constraints and outlines potential directions for future research to extend and strengthen the findings.
Point 1: The English should be polished to improve clarity, grammar, and flow.
Response 7: The proof reading of the manuscript was done thoroughly in order to improve the language clarity, grammar, and consistency of the technical aspects.
Point 2: Correct spelling (for example, “insurance”).
Use one consistent term: “Vertical Federated Learning (VFL).”
Use one name for risk_score/risc_score and define it once.
Response 2: Agree. I/We have, accordingly, revised this point
The word "Insurrance" was amended to the word "Insurance" the word risk score was also corrected to risc_score and the acronym (Vertical Federated Learning (VFL)) was used uniformly in the manuscript.

Reviewer 3 Report
Comments and Suggestions for Authors
The manuscript is much improved in most accounts, however there are points mentioned in the comments file which are not visible in the manuscript. For instance, the response to comment 2 is not in the text. We require the actual clear contributions to be present in both the abstract and the latter part of the introduction, as this is needed for the reader to know precisely what the value is. The best way to do this is to state and enumerate the contributions at the end of the Introduction in a crystal-clear manner.
Also with reference to comment 7, the authors did not entirely understand the reviewer note. This is not about the research article format. The issue is that Sections 3.7 to 3.10 overall remain largely unchanged and still have the form of a very fragmented series of statements, where qualification is absent and it remains unclear how and why this series of sections contribute to clarity. This part is yet to be updated.
Author Response
Comment 1: We require the actual clear contributions to be present in both the abstract and the latter part of the introduction, as this is needed for the reader to know precisely what the value is. The best way to do this is to state and enumerate the contributions at the end of the Introduction in a crystal-clear manner
Response 1: Thank you for pointing this out. I/We agree with this comment.
According to the Abstract, the federated configuration [with two entities, 10 rounds, ε = 1.0] is more suitable; furthermore, the last paragraph of the Introduction has been revised to make a new contribution part that lists main achievements as follows (i) privacy-preserving VFL-based fraud detection, (ii) Gaussian-noise-based Differential Privacy, and hybrid FFNN Transformer model, and (iii) over 99 percent federated learning accuracy.
Comment 2: The issue is that Sections 3.7 to 3.10 overall remain largely unchanged and still have the form of a very fragmented series of statements, where qualification is absent and it remains unclear how and why this series of sections contribute to clarity. This part is yet to be updated.
Response 2: Agree. I/We have, accordingly, revised.
Sentences that connect Sections 3.7 to 3.10 were revised. This revision helps readers readily grasp how each subtopic adds value(e.g., achieving groups, decisions that are both secure and fraud detection VFL).
Point 1: Language, consistency, and clarity require further improvement.
Response 1: The entire manuscript was carefully proofread and edited for flow and academic style format, and for compliance with MDPI standards. Spelling, grammar and formatting were corrected where necessary and transitions to sections polished for better reading.
